# Multi-Trait Genome-Wide Association Studies of *Sorghum bicolor* Regarding Resistance to Anthracnose, Downy Mildew, Grain Mold and Head Smut

**DOI:** 10.3390/pathogens12060779

**Published:** 2023-05-30

**Authors:** Ezekiel Ahn, Louis K. Prom, Clint Magill

**Affiliations:** 1USDA-ARS Plant Science Research Unit, St. Paul, MN 55108, USA; 2USDA-ARS Southern Plains Agricultural Research Center, College Station, TX 77845, USA; louis.prom@usda.gov; 3Department of Plant Pathology and Microbiology, Texas A&M University, College Station, TX 77843, USA

**Keywords:** *Sorghum bicolor*, fungal pathogens, multivariate GWAS, principal component analysis

## Abstract

Multivariate linear mixed models (mvLMMs) are widely applied for genome-wide association studies (GWAS) to detect genetic variants affecting multiple traits with correlations and/or different plant growth stages. Subsets of multiple sorghum populations, including the Sorghum Association Panel (SAP), the Sorghum Mini Core Collection and the Senegalese sorghum population, have been screened against various sorghum diseases such as anthracnose, downy mildew, grain mold and head smut. Still, these studies were generally performed in a univariate framework. In this study, we performed GWAS based on the principal components of defense-related multi-traits against the fungal diseases, identifying new potential SNPs (S04_51771351, S02_66200847, S09_47938177, S08_7370058, S03_72625166, S07_17951013, S04_66666642 and S08_51886715) associated with sorghum’s defense against these diseases.

## 1. Introduction

Sorghum (*Sorghum bicolor* (L.) Moench), a multipurpose crop used as a source of food, fodder, feed and fuel, is ranked among the top five cereal crops worldwide [1]. Still, the yields are highly constrained due to fungal pathogens that cause severe diseases in the crop. Among various fungal diseases, anthracnose caused by *Colletotrichum sublineola* Henn. ex Sacc. and Trotter 1913 is one of sorghum’s most destructive fungal diseases, with annual yield losses of up to 100% [2]. Downy mildew, caused by *Peronosclerospora sorghi*, is another critical disease of sorghum that can cause severe epidemics, resulting in considerable yield losses [3]. Head smut, caused by the soil-borne facultative biotrophic basidiomycete *Sporisorium reilianum* (Kühn) Langdon and Fullerton (syns. *Sphacelotheca reiliana* (Kühn) G.P. Clinton and *Sorosporium reilianum* (Kühn) McAlpine), is a serious global sorghum disease as well [4,5]. Grain mold, a complex fungal disease, is associated with many fungal species, such as *Fusarium* spp., *Curvularia lunata* (Wakker) Boedijn, *Alternaria alternata* (Fr.) Keissl. (1912) and *Phoma sorghina* (Sacc.). Boerema, Dorenb. and Kesteren is considered one of the most important biotic constraints of grain sorghum production worldwide [3].

Sorghum’s resistance to diseases caused by fungal pathogens heavily relies on novel sources of resistance genes or defense response genes identified by genome-wide association studies (GWAS) that test hundreds of thousands of genetic variants across many genomes to find those statistically associated with a specific trait or disease [6]. In GWAS, linear mixed models (LMMs or MLMs) have been extensively applied, and most GWAS are conducted under a univariate framework. Still, recent studies began to apply the multivariate linear mixed model (mvLMM) to GWAS due to the increased importance of detecting genetic variants that affect multiple traits or different growth stages [7]. Principal component analysis (PCA) is a valuable tool that has been widely used for the multivariate analysis of correlated variables, including mvLMM, and it has been shown that multivariate GWAS approaches yield a higher true-positive quantitative trait nucleotide (QTN) detection rate than comparable univariate approaches [8,9]. 

In sorghum pathology, multiple subsets of sorghum populations, including the Sorghum Association Panel (SAP), the Sorghum Mini Core Collection and the Senegalese sorghum population, have been screened against various sorghum diseases such as anthracnose, downy mildew, grain mold and head smut. Still, these studies were performed in a univariate framework [10,11,12,13,14,15,16,17]. The studies measured multiple disease-related traits in the three different populations, and both single-nucleotide polymorphism (SNP) data and phenotypic data are publicly available. In this study, a multivariate GWAS was conducted based on the top principal components (PC GWAS with LMMs) of defense-related traits against fungal pathogens, resulting in the identification of new potential SNPs associated with sorghum’s defense against these diseases.

## 2. Materials and Methods

### 2.1. Sorghum Association Panel

Prom et al. [12,17] evaluated 377 lines from the Sorghum Association Panel (SAP) for grain mold resistance by inoculating them with either *A. alternata* alone; a mixture of *A. alternata*, *Fusarium thapsinum* and *Curvularia lunata*; or untreated water as a control group at the Texas A&M AgriLife Research Farm, Burleson County, Texas. Based on the phenotypic data, a univariate GWAS with over 79,000 SNP loci from a publicly available genotype was conducted by sequencing dataset for the SAP lines [18] using the TASSEL version 5.2.55 [19] association mapping software to identify chromosomal locations associated with differences in the grain mold response. Of the three treatments, a PCA was performed with TASSEL version 5.2.88 [19], and a multivariate GWAS (PC GWAS based on PCs (PC1 = data and PC2&3 = covariates)) was conducted through LMMs (Appendix A). False associations were reduced by removing SNPs with greater than 20% unknown alleles and SNPs with minor allele frequency (MAF) below 5% [20].

### 2.2. Sorghum Mini Core Collection

Ahn et al. [13] performed a univariate GWAS based on phenotypic data of 242 Mini Core lines which had been screened for anthracnose, downy mildew and head smut resistance. The screening results were combined with over 290,000 SNP loci from an updated version of a publicly available genotype by a sequencing (GBS) dataset available for the Mini Core collection [21,22,23,24], and the GAPIT (Genome Association and Prediction Integrated Tool) R package was used to identify chromosomal locations that differed in their disease responses [13,25,26]. With the phenotypic data of the four pathogens, a PCA was performed with TASSEL version 5.2.88 (Appendix A) [19], and a multivariate GWAS (PC GWAS based on PC1-3 (PC1 = data and PC2&3 = covariates)) was performed through LMMs by applying the identical standard to filter false associations out [20]. 

### 2.3. Senegalese Sorghum Population

A total of 159 to 163 Senegalese sorghum lines were evaluated for their responses to *C. sublineola* (traits included average score for seedling inoculation, highest score for seedling inoculation, and average score for 8-leaf-stage sorghum inoculation) and *S. reilianum* (traits included the appearance rate of dark spots in seedlings and the average time of the spot appearance) [10,11,14]. A total of 193,727 SNP data points were extracted from an integrated sorghum SNP dataset based on sorghum reference genome version 3.1.1, and were originally genotyped using GBS [21,22,23,24]. PCA combined the three traits for *C. sublineola* (average score for seedling and 8-leaf-stage inoculations and the highest score for seeding inoculation), and the two traits (spot appearance rate and time) were used to generate PC 1&2 for *S. reilianum* through TASSEL version 5.2.88 [19]. Furthermore, all five traits were analyzed with PCA to identify potential plant defense-related loci commonly associated with the two diseases. Multivariate GWAS (PC GWAS based on PC1 = data and other PCs = covariates)) was performed using the method described above (Appendix A). 

The SNPs that passed the Bonferroni correction were mapped back to the published sorghum reference genome to be tracked to the specific chromosome location based on the sorghum reference genome sequence, version 3.1.1, available at the Phytozome 13 (https://phytozome.jgi.doe.gov accessed on 5 March 2023) [27], and the top SNPs that failed to pass Bonferroni correction were also checked to identify the nearest genes from SNPs that had previously reported roles in biotic or abiotic resistance/stress responses.

## 3. Results

Multivariate GWAS identified nine SNPs that passed the Bonferroni threshold (Figure 1 and Table 1). Among the nine SNPs, only the SNP locus S01_72868925 had previously been reported as a top SNP in univariate GWAS studies [10]; the other eight SNPs were novel sources of resistance to fungal pathogens in sorghum.

## 4. Discussion

The subsets of the three populations used in this study were thoroughly analyzed for population structure, genetic analysis, and linkage disequilibrium (LD) in recent studies [14,18,21,22,24]. Sorghum pathologists have been applying GWAS to identify defense-related genes in sorghum against various fungal diseases, and the studies successfully mined various sources of sorghum resistance in univariate frameworks [10,11,12,13,15,20,28]. Examples of candidate defense-related genes through univariate GWAS include F-box domain, helix–loop–helix DNA-binding domain, K05280-flavonoid 3’-monooxygenase, leucine-rich repeat, MADS-box protein, MYB transcription factor, oryzalide A biosynthesis, pentatricopeptide repeat, selenium binding protein and xyloglucan endotransglucosylase [10,11,12,13,14,15,16,17,28]. In addition to univariate GWAS, recent studies have extensively applied multivariate GWAS, identifying new sources of SNPs associated with various traits of interest. PCA is often applied to conduct multivariate GWAS [8,9]. 

In this study, we revisited multiple studies that applied univariate GWAS in experiments on sorghum pathology. We reanalyzed the data in multivariate frameworks, obtaining eight novel SNPs potentially associated with sorghum’s defense against fungal pathogens (Figure 1 and Table 1). Sobic.005G141700 was identified as a top candidate for grain mold resistance under different inoculations in the SAP lines. Sobic.005G141700 is speculated to be associated with D-mannose binding lectin. Although not an identical molecular marker, one study reported that the SNP locus S02_61590648, a coding region similar to the mannose-binding lectin coding region of pepper, was a top candidate for sorghum defense against downy mildew [13], and mannose-binding lectins are essential for plant defense signaling during pathogen attacks as they recognize specific carbohydrates on pathogen surfaces [29].

In the Sorghum Mini Core Collection, the SNP locus S04_51771351, which passed the Bonferroni threshold, was only 381bp away from the zinc finger-related gene (Sobic.004G167500). Zinc finger proteins have been shown to play diverse roles in plant stress responses [30], and multiple previous studies have identified zinc finger-related proteins for sorghum defense-related genes against fungal pathogens [12,13,15]. 

Nine SNP loci associated with anthracnose resistance passed the Bonferroni threshold in the Senegalese sorghum lines. Except for the SNP locus S01_72868925 (tagged to Sobic.001G451800), a reported SNP for potential anthracnose resistance in sorghum seedlings associated with leucine-rich repeat [10], the other eight SNPs were newly identified and had not previously been reported as top SNPs in univariate GWAS studies. The SNP locus S02_66200847 is closely located to Sobic.002G280800, associated with the cAMP response element binding protein, which has remarkable roles in plant defense, salt stress, and ethylene responses [31]. The SNP locus S09_47938177 is associated with Sobic.009G126000, which encodes for endo-beta-mannosidase. This gene regulates sugar metabolism and cell wall function [32,33,34]. 

The SNP locus S08_7370058, located 1117 bp away from Sobic.008G065700, is associated with splicing factors. Splicing factors function in various physiological processes, including plant disease resistance and abiotic stress responses [35,36]. The SNP locus S03_72625166 tagged Sobic.003G421201, a gene related to hexokinase-3. In *Nicotiana benthamiana* Domin, hexokinases are known to play a role in controlling programmed cell death [37]. Sobic.007G096700 tagged by SNP locus S07_17951013 is similar to Ole e 1 proteins involved in pollen tube development [38]. It is unclear whether the gene is associated with sorghum defense or if it is simply a false positive. If the SNP is a false positive, the SNPs exhibiting high LD with SNP locus S07_17951013 in Table 1 may potentially be truly associated with sorghum defense response.

The SNP locus S04_66666642, located in the glucose-related gene Sobic.004G334300, was also listed as a top candidate. Numerous studies have shown that sugars play a crucial role in plant defense responses to various abiotic and biotic stress factors [39]. The MYB transcription factor (Sobic.008G112200) was tagged by the SNP locus S08_51886715, and MYB transcription factors are reported to be associated with sorghum defense against fungal diseases such as smut and head smut [11,40]. Although no SNPs passed the Bonferroni threshold, PC GWAS based on the traits associated with head smut and combinations of the traits related to anthracnose and head smut in the Senegalese sorghum lines identified a few SNPs of potential importance. 

Although not the same SNPs, zinc finger- (Sobic.008G125400) and calcium-related binding proteins (Sobic.002G113900) have been identified as top candidates conferring defense in plants in several GWAS studies [14]. Another SNP locus, S05_8867065, tagged Sobic.005G073200, which showed similarity to the DUF803 domain. This domain was reported to be one of the top candidate SNPs that differed between resistant and susceptible types of Upland cotton (*Gossypium hirsutum* L.) against the southern root-knot nematode [*Meloidogyne incognita* (Kofoid & White, 1919)] [41]. Lastly, the SNP locus S03_9787536 tagged DUF3339 (Sobic.003G108200). Although not in an antagonistic relationship, mycorrhizal symbiosis upregulated DUF3339 in *Vaccinium myrtillus* L. 1753 [42]. Among the top SNPs identified in this study, the SNP locus S01_72868925 was previously reported as a top SNP in univariate GWAS studies [10]. Still, other SNPs were newly listed, indicating the exceptional potential of multivariate GWAS for sorghum–pathogen interactions.

Nearly all the candidate genes identified in this study through multivariate GWAS had known defense functions in plants. The genes’ functions can be further analyzed and validated through advanced techniques such as real-time quantitative reverse transcription PCR (Real-time qRT-PCR), RNA sequencing analysis (RNA-Seq) and CRISPR-Cas9-associated gene editing. However, validating these functions requires tremendous time and funding. Agrobacterium-mediated transformation and regeneration protocols in monocot crops, such as maize and sorghum, are notoriously difficult to transform, with complications arising in both methodologies for each of these crops and long tissue culture periods being required for the regeneration of plants from transformed tissue [43].

Hence, it is essential to continue mining novel SNPs associated with sorghum defense against fungal pathogens in order to obtain more precise knowledge on crucial sorghum defense genes before functional validations of candidate genes are carried out. Applying multi-trait GWAS in sorghum pathology will provide avenues to identify further defense-related sorghum genes that have yet to be revealed. Additionally, conducting multivariate GWAS on multiple fungal pathogens of sorghum could reveal defense-related genes that are either pathogen-specific or universally involved in sorghum defense.

## Figures and Tables

**Figure 1 pathogens-12-00779-f001:**
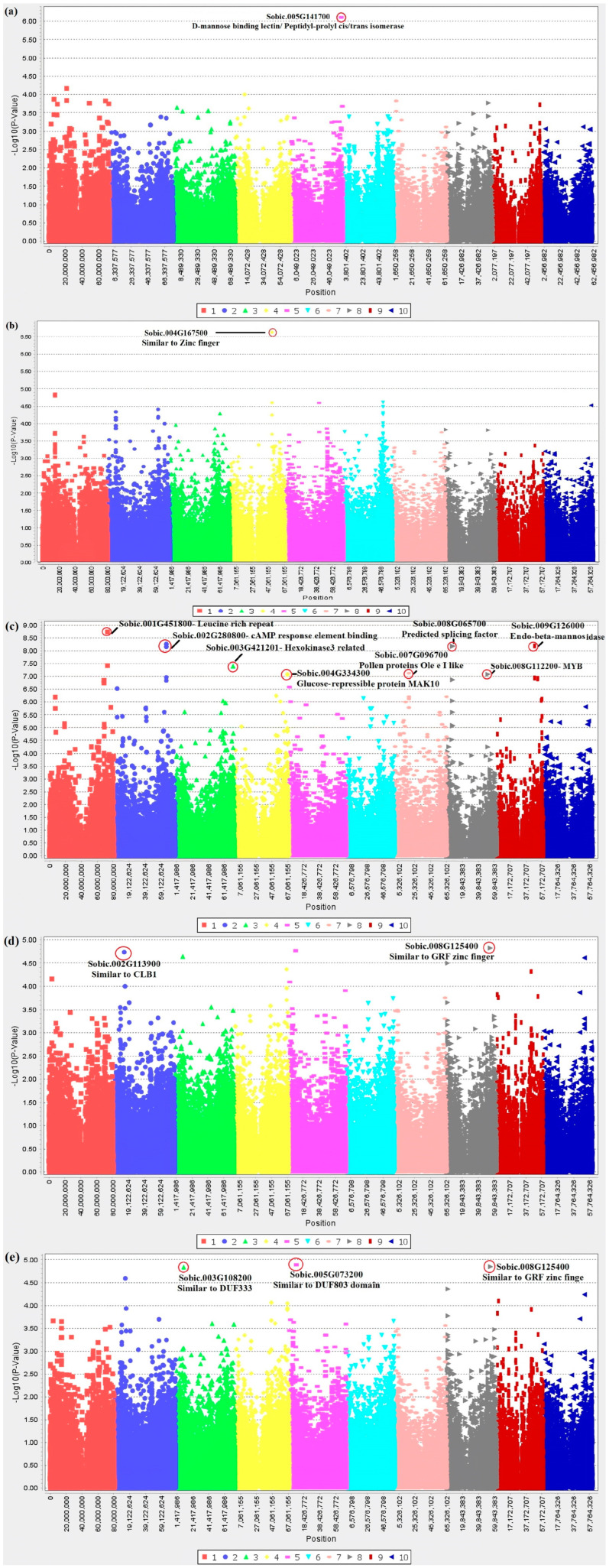
The multivariate genome-wide association for response to multiple fungal pathogens. Manhattan plots based on PC GWAS demonstrate the top candidate SNPs. (**a**) The Manhattan plot of SAP for multivariate GWAS for grain mold resistance under three conditions. (**b**) Manhattan plot of the Mini Core collection for multivariate GWAS for combinatory PCs of anthracnose, downy mildew and head smut. The top SNP passed the Bonferroni threshold. (**c**) Manhattan plot of Senegalese sorghum germplasms for anthracnose, with three multi-traits. Eight SNPs passed the Bonferroni threshold. (**d**) Manhattan plot of Senegalese sorghum germplasms for head smut with two traits. (**e**) Manhattan plot of Senegalese sorghum germplasms for anthracnose and head smut-related traits. The top SNPs are highlighted with red circles, and annotated functions are described near the SNPs.

**Table 1 pathogens-12-00779-t001:** Annotated genes shown nearest to the most significant SNPs were associated with multi-trait GWAS in all three sorghum populations. The distance, in base pairs, to the nearest genes and *p*-value are shown. * = passed Bonferroni correction.

Population	Traits	Chr	Location	Candidate Gene and Function	Base Pairs	SNP Base %	TASSEL *p*-Value	SNPs with High Linkage Disequilibrium (LD) (0.7 < R^2^), Closest Gene and Function
SAP	Grain mold under three treatments	5	60278659	Sobic.005G141700 Uncharacterized Associated PlantFAMs- D-mannose binding lectin/Peptidyl-prolyl cis/trans isomerase	3147	A: 13.1% C: 86.9%	0.000000774 Bonferroni ≈ 0.00000041	S05_60274552 S05_60274571 **Nearest gene** Sobic.005G141700
Mini Core	Anthracnose + Downy mildew + Head smut	4	51771351	Sobic.004G167500 Uncharacterized Functional annotation- Zinc finger, RING/FYVE/PHD-type	381	C: 79.9% T: 20.1%	0.000000236 * Bonferroni ≈ 0.0000003	None
Senegalese	Anthracnose with three variates	1	72868925 and multiple SNPs nearby	Sobic.001G451800 Protein kinase domain//Leucine-rich repeat N-terminal domain	0	G: 79.9% T: 20.1%	0.00000000188 * Bonferroni ≈ 0.00000017	S01_72831422 **Nearest gene** Sobic.001G451400- Vacuole morphology and inheritance protein 14
		2	66200847 and multiple SNPs nearby	Sobic.002G280800 cAMP response element binding protein (CREB) Associated PlantFAMs- bZIP transcription factor domain	2920	C: 20.1% G: 79.9%	0.00000000553 * Bonferroni ≈ 0.00000017	S02_66173353 **Nearest gene** Sobic.002G280300 PF05553–Cotton fiber expressed protein (DUF761) S02_66179471 S02_66179497 **Nearest gene** Sobic.002G280400 Acyl carrier protein /Zinc finger protein 593-related S02_66186569 S02_66188204 S02_66188632 S02_66188704 S02_66188732 **Nearest gene** Sobic.002G280600 Mediator of RNA polyerase II transcription subunit 4 S02_66189758 S02_66189765 S02_66189995 S02_66190001 S02_66190035 S02_66194427 S02_66194452 S02_66196956 S02_66196986 **Nearest gene** Sobic.002G280700 Histone-binding protein RBBP4 S02_66200592 S02_66200825 **Nearest gene** Sobic.002G280800 Camp-response element binding protein-related
		9	47938177 and multiple SNPs nearby	Sobic.009G126000 Mannosyl glycoprotein endo-beta-mannosidase	0	A: 17.0% G: 83%	0.00000000645 * Bonferroni ≈ 0.00000017	None
		8	7370058 and multiple SNPs nearby	Sobic.008G065700 Uncharacterized Associated PlantFAMs- Predicted splicing factor	1117	A: 22.8% C: 77.2%	0.0000000067 * Bonferroni ≈ 0.00000017	S08_7327804 S08_7345722 **Nearest gene** Sobic.008G065600 Auxin responsive GH3 gene family S08_7363922 **Nearest gene** Sobic.008G065700
		3	72625166	Sobic.003G421201 Hexokinase-3 related	0	A: 21.0% G: 79.0%	0.0000000404 * Bonferroni ≈ 0.00000017	S03_72617056 S03_72617102 **Nearest gene** Sobic.003G421100 Gulonolactone oxidase
		7	17951013	Sobic.007G096700 Pollen proteins Ole e I like	0	G: 85.5% T: 14.5%	0.0000000752 * Bonferroni ≈ 0.00000017	S07_17760523 S07_17760525 S07_17760589 S07_17761449 S07_17761552 **Nearest gene** Sobic.007G096000 Enhancer of polycomb S07_17831891 S07_17831935 **Nearest gene** Sobic.007G096200 Nucleoporin nup43 S07_17882047 S07_17882072 S07_17882080 **Nearest gene** Sobic.007G096401 Uncharacterized
		4	66666642 and multiple SNPs nearby	Sobic.004G334300 Similar to Glucose-repressible protein MAK10	0	A: 32.5% G: 67.5%	0.0000000813 * Bonferroni ≈ 0.00000017	S04_66651495 **Nearest gene** Monogalactosyldiacylglycerol synthase 2
		8	51886715	Sobic.008G112200 MYB transcription factor	0	C: 91.2% T: 8.8%	0.0000000834 * Bonferroni ≈ 0.00000017	S08_51822410 S08_51822572 **Nearest gene** Sobic.008G112000 Oxidoreductase, 2OG-FE II S08_51868277 S08_51868331 **Nearest gene** Sobic.008G112100 Lysophosphatidic acid acyltransferase / lysophosphatidylinositol acyltransferase S08_51886688 S08_51886691 **Nearest gene** Sobic.008G112200
	Head smut with two variates	8	54897663	Sobic.008G125400 Uncharacterized Associated PlantFAMs- GRF zinc finger (zf-GRF)	0	C: 70.8% G: 29.2%	0.000015 Bonferroni ≈ 0.00000017	None
		2	13906340	Sobic.002G113900 Uncharacterized Similar to calcium-dependent lipid binding (CLB1) protein	0	A: 67.0% G: 33.0%	0.0000184 Bonferroni ≈ 0.00000017	None
	Anthracnose + Head smut	5	8867065	Sobic.005G073200 Uncharacterized Associated PlantFAMs- DUF803 domain containing	0	A: 37.8% T: 62.2%	0.000013 Bonferroni ≈ 0.00000017	None
		8	54897663	Sobic.008G125400 Uncharacterized Associated PlantFAMs- GRF zinc finger (zf-GRF)	0	C: 70.8% G: 29.2%	0.0000143 Bonferroni ≈ 0.00000017	Identical SNP above
		3	9787536	Sobic.003G108200 Uncharacterized Associated PlantFAMs- protein of unknown function (DUF3339)	0	C: 48.6% T: 51.4%	0.0000146 Bonferroni ≈ 0.00000017	None

## Data Availability

Not applicable.

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
