# Peer review of "Multi-Trait Genome-Wide Association Studies of Sorghum bicolor Regarding Resistance to Anthracnose, Downy Mildew, Grain Mold and Head Smut"

_pathogens, 2023, doi:10.3390/pathogens12060779_

Round 1

Reviewer 1 Report

The GWAS-based study reported by Ahn et al. is interesting. However, this is more of a preliminary study where only the identification of potential SNPs has been carried out. Although the outcomes are welcome, but no validation works has been envisaged in this study.

Author Response

We understand the reviewers' concern, and gene validations through Crispr/Cas9 are essential. However, despite the development of WUS and BBM through Corteva, sorghum gene editing, and transformation can be years of work for the genes we listed. Furthermore, we need lots of equipment and space to do a successful transformation, such as a tissue culture room and incubators. The purpose of this manuscript is to suggest more candidates, so other researchers who can do validation studies will have more options to explore the list of genes they want to work with.  It is unfortunate, but as the funding for the project is already expired, we can't conduct CRISPR/Cas9-based gene edits for functional validation. Still, we extensively revised the manuscript by providing potential SNPs linked with our candidate SNPs to broaden the study.

Reviewer 2 Report

This study applied multivariate linear mixed models (mvLMM) for genome-wide association studies (GWAS) to identify genetic variants that affect multiple traits in different growth stages in plants. Specifically, the study focuses on screening multiple sorghum populations against various fungal pathogens using defense-related multi-traits, and identifies several potential SNPs associated with sorghum's defense against these pathogens.

The use of mvLMM for GWAS is a powerful approach that accounts for correlations between multiple traits and different growth stages, providing a more comprehensive understanding of the genetic architecture of complex traits. The study's focus on sorghum, a major cereal crop in many regions, has practical implications for improving sorghum's resistance to fungal pathogens that can cause significant yield losses.

However, it is important to note that further validation and functional analysis are needed to confirm the potential SNPs' roles in sorghum's defense against fungal pathogens.

Please consider the following comments:

The method and result sections are underdeveloped. To mine candidate genes resistance to pathogens, haplotype block estimation is strongly recommended to be performed.

Moreover, the results of population structure, genetic analysis, and linkage disequilibrium analysis should be shown clearly. (There are some GWAS papers can be referred such as https://doi.org/10.1186/s12870-020-02603-0 and https://doi.org/10.1186/s12870-021-03146-8)

Author Response

Reviewer: It is important to note that further validation and functional analysis are needed to confirm the potential SNPs' roles in sorghum's defense against fungal pathogens.

Our response: We understand the reviewers' concern, and gene validations through Crispr/Cas9 are essential. However, despite the development of WUS and BBM through Corteva, sorghum gene editing, and transformation can be years of work for the genes we listed. Furthermore, we need lots of equipment and space to transform successfully, such as a tissue culture room and incubators. The purpose of this manuscript is to suggest more candidates, so other researchers who can do validation studies will have more options to explore the list of genes they want to work with.  It is unfortunate, but as the funding for the project is already expired, we can't conduct CRISPR/Cas9-based gene edits for functional validation.

Reviewer: To mine candidate genes' resistance to pathogens, haplotype block estimation is strongly recommended to be performed.

Our response: Haplotype block estimation is an excellent way to extend potential candidate SNPs associated with the resistance genes, but it often gives too broad a spectrum of genes.  Instead, we extensively revised the manuscript by providing potential SNPs linked with our candidate SNPs with high LD scores (R square value<0.7) to broaden the study. In this way, we narrowed our potential SNPs being missed by simply providing the nearest genes to our top candidate SNPs. 

Reviewer: Moreover, the results of population structure, genetic analysis, and linkage disequilibrium analysis should be shown clearly.

Our response: These analyses were conducted in Genome-wide scales in the original studies we used for SNP data.  Although we can do it again with small-scale in our subset of the Senegalese line, we don't think it is any better than the original studies. We cited the works in the manuscript to clarify the matters. 

Round 2

Reviewer 1 Report

The authors have significantly improved the manuscript in the revision. However, I hope the authors should try to validate some of the identified candidates in the study in future.